# A Proposed Interpretation of the Wave–Particle Duality

**DOI:** 10.3390/e24111535

**Published:** 2022-10-26

**Authors:** Kurt Jung

**Affiliations:** Fachbereich Physik, Universität Kaiserslautern, 67663 Kaiserslautern, Germany; jung@physik.uni-kl.de

**Keywords:** foundations of quantum mechanics, wave–particle duality, escort wave concept, vacuum fluctuations, matter wave, phase coupling, classical trajectories, guiding wave, Lorentz covariance

## Abstract

Within the framework of quantum mechanics, the wave function squared describes the probability density of particles. In this article, another description of the wave function is given which embeds quantum mechanics into the traditional fields of physics, thus making new interpretations dispensable. The new concept is based on the idea that each microscopic particle with non-vanishing rest mass is accompanied by a matter wave, which is formed by adjusting the phases of the vacuum fluctuations in the vicinity of the vibrating particle. The vibrations of the particle and wave are phase-coupled. Particles move on continuous approximately classical trajectories. By the phase coupling mechanism, the particle transfers the information on its kinematics and thus also on the external potential to the wave. The space dependence of the escorting wave turns out to be equal to the wave function. The new concept fundamentally differs from the pilot wave concept of Bohmian mechanics.

## 1. Introduction

Shortly after the development of modern quantum mechanics by Werner Heisenberg [1] and Erwin Schrödinger [2,3,4,5,6], Max Born [7,8,9,10] proposed that the absolute square of the wave function is equal to the probability of finding the associated particle at this location. At the beginning, Born formulated the probability wave hypothesis only for scattering processes. Subsequently, the scope of the hypothesis was extended to bound states.

Already in 1924, Louis de Broglie [11] presented in their doctor thesis a relativistic model of particles and coexisting waves. De Broglie’s phase wave has essentially the same properties as the escort wave presented in this article. However, de Broglie did not account for the stochastic distortion of the phase wave and thus also of the particle’s trajectory by the continuous interaction with vacuum fluctuations. The phase wave concept would have been an excellent basis for further considerations. Unfortunately, de Broglie’s pilot wave theory published in 1927 [12] was a step backwards because the structure of the pilot wave was no longer related to the kinematics of the particle. Only decades later, de Broglie vainly suggested a double solution model where he also introduced stochastic modification of the trajectories [13,14,15,16].

In 1952, David Bohm [17,18] proposed a dual theory with particles and waves as concrete physical objects. This theory is known as de Broglie–Bohm theory or as Bohmian mechanics. The escort wave concept, which will be presented in this article, and Bohmian mechanics fundamentally differ from each other although both theories assume the coexistence of particles and waves. The differences will be discussed near the end of this article when the escort wave concept has been explained in detail.

In the last one hundred years, countless concepts have been published in order to explain quantum phenomena. Nearly all combinations of imaginable ideas have been proposed. Some of these ideas are very promising. Until now, it was not possible to find a complete solution that was very convincing. For the purpose of better understanding the new approach, numerous well-known concepts, usually treated in quantum mechanical text books, also have to be explicitly mentioned in this article.

## 2. Fundamental Considerations on the Wave–Particle System

In microphysics, particles exhibit a Janus-faced character. They are strictly localized in space, but their propagation complies with wave laws. This ambivalence becomes obvious when particles are diffracted at a double slit aperture. Each particle causes a single spot on a photographic plate. If the experiment is repeated very often, the pattern on the photographic plate approaches the results of diffraction experiments with waves.

Although a strictly localized particle can only pass through one of the slits, the probability distribution conforms to the circumstance that both slits are open. There is no interaction between individual particles because the diffraction pattern remains unchanged if the experiment is performed with such small beam intensities that only single particles are traveling between the source and detection plane. Apparently, even single particles obey wave rules. This means that each particle has to be accompanied by an extended physically real wave which provides the particle with the information on the second slit. Otherwise, particles do not obtain the information that they need to avoid regions where the partial waves going out from the two slits mutually extinguish themselves by destructive interference.

The Copenhagen interpretation of quantum mechanics starts from the premise that the behavior of particles with non-zero rest mass is controlled by a probability wave. This probability wave is assumed to be a purely mathematical tool without any physical substance. In this article, it will be shown that it is appropriate to assume that every particle with non-zero rest mass is accompanied by a real wave. The escorting wave emerges from the interplay of massive particles with fluctuating matter waves ubiquitously present in vacuum due to the continual creation and annihilation of short-living particles.

## 3. Emergence of the De Broglie Wave Length

The wave four-vector
(1)$$K^{\mu }=\left(\omega /c,k,0,0\right)$$
characterizes a plane wave with respect to its angular frequency ω and its wave number *k*. For matter waves accompanying massive particles, the norm squared of the wave four-vector is non-zero. It can be written in the form
(2)$$K^{\mu }K_{\mu }=\omega^{2}/c^{2}-k^{2}=\Omega^{2}/c^{2}$$
with the rest or minimum angular frequency Ω. The rest frequency Ω is coupled to the rest mass of the particle *m* by the relation
(3)$$mc^{2}=\hbar \Omega .$$
This relationship arises as a result when the momentum four-vectors of particle and wave are compared.

The angular frequency of a matter wave is given by
(4)$$\omega =\sqrt{k^{2}c^{2}+\Omega^{2}}.$$
Thus, the phase velocity is
(5)$$v_{p}=\frac{\omega }{k}=\sqrt{c^{2}+\Omega^{2}/k^{2}}>c$$
and the group velocity is
(6)$$v_{g}=\frac{d\omega }{dk}=\frac{2kc^{2}}{2\sqrt{k^{2}c^{2}+\Omega^{2}}}=\frac{kc^{2}}{\omega }=\frac{c^{2}}{v_{p}}<c.$$
The group velocity denotes the velocity of a propagating wave packet and thus also the maximum velocity of a signal transported by the matter wave. In accordance with special relativity, the group velocity of matter waves is always smaller than the speed of light.

In the following, the laboratory system *P* and the system $P^{\prime }$ are used. $P^{\prime }$ is moving with the velocity v with respect to the laboratory system *P*. Without loss of generality, the considerations will be restricted to the case in which the velocity v=(v,0,0) is oriented in parallel to the *x* axis.

Figure 1 shows the coordinate systems of the two reference frames *P* and $P^{\prime }$. The origins of the two coordinate systems are chosen to be equal for $t=t^{\prime }=0$. Traditionally, the ct axis of the laboratory system *P* is plotted in the vertical direction and the *x* axis is plotted in the horizontal direction. All events on the *x* axis are simultaneous in the laboratory system.

The axes $ct^{\prime }$ and $x^{\prime }$ of the system $P^{\prime }$ are both tilted by the angle α towards the bisecting plane ct=x. The $ct^{\prime }$ axis is described by the equation ct=xc/v and the $x^{\prime }$ axis is described by the equation ct=xv/c. This means that the two axes have reciprocal slopes. All events on the $x^{\prime }$ plane are simultaneous in the system $P^{\prime }$. The relativistic effect of the planes of simultaneity being tilted with respect to one another must already be taken into account for low velocities because the leading term of the power series expansion is proportional to v/c.

The $x^{\prime }$ axis and the two lines parallel to the $x^{\prime }$ axis represent three wave fronts of a plane matter wave. The two lines above and below the $x^{\prime }$ axis are described by
(7)$$ct=x\frac{v}{c}+c\tau^{*}\;\mathrm{and}\;ct=x\frac{v}{c}-c\tau^{*}$$
where $\tau^{*}$ denotes the wave period measured in the laboratory system. As the phases of adjacent wave fronts differ by a wave period, the oscillations at all locations of the three wave fronts are in phase.

The de Broglie wave length of the escort wave is given by the distance of the crossings of two consecutive wave fronts with the *x* axis. As the middle phase line crosses the *x* axis at the origin, the de Broglie wave length is given by the intersection of the lower phase line with the *x* axis. If one inserts the constraint ct=0 into the formula for the lower wave front, one obtains the x coordinate of the intercept of x axis and the wave front
(8)$$\lambda =c\tau^{*}\frac{c}{v}.$$
This relation can be easily checked because the ct axis as well as the *x* axis are subdivided in units of $c\tau^{*}$. In Figure 1, the velocity *v* was chosen to be v=c/6. Therefore, the de Broglie wave length measured in the laboratory system amounts to $6\;\;c\tau^{*}$. The geometrical derivation of the de Broglie wave length clearly shows that the de Broglie wave length emerges from the relativistic tilting of the simultaneity plane.

## 4. Conformance of the Kinematics of Particle and Wave

The rest systems of a particle and the matter wave accompanying it agree with each other. As the matter wave has a limited coherence length, it has the character of a wave packet. The particle is usually located in the center of the wave packet. The feature that the particle is surrounded by the wave is a scalar property. Thus, this attribute holds in all frames of reference.

By using Equation (Equation 3), the de Broglie wave length given in Equation (Equation 8) can be transformed into the well-known expression
(9)$$\lambda =c\tau^{*}\frac{c}{v}=\frac{2\pi }{\gamma \Omega }\frac{c^{2}}{v}=\frac{2\pi \hbar }{\gamma \hbar \Omega }\frac{c^{2}}{v}=\frac{h}{\gamma mc^{2}}\frac{c^{2}}{v}=\frac{h}{\gamma mv}$$
with the frequency γΩ of the matter wave measured in the laboratory system.

If the particle is smoothly accelerated, the wave accompanying it is also accelerated. Thus, even under the influence of varying external potentials, the momenta of the particle and accompanying wave packet continue to be equal. The particle and wave packet follow the same trajectory. Each particle is permanently escorted by an extended wave with a quite large but limited coherence length. Locally, trajectories and wave fronts always have reciprocal slopes.

An escorting wave must not only contain the information on the velocity of the particle but also of the higher derivatives of the trajectory. In fact, the potential must be infinitely differentiable. The information on the higher derivatives can be incorporated into the wave because the wave is extended in space. Thus, the de Broglie wave length changes within the wave packet if the particle is accelerated. The flux continuity condition leads to a variation of the wave amplitude. At each location, the particle and wave have the same kinetic energy. Traveling on its orbit, the particle will generally be far from the turning points. Nonetheless, the wave will be reflected at these points because its necessarily non-negative kinetic energy reaches the zero line.

The particle and escorting wave move in unison until an abrupt acceleration—for example, due to a collision with another particle—breaks up the close phase coupling. Subsequently, the mutual influence is suspended. The wave immediately loses its guiding role and is no longer stabilized by the particle. Therefore, such a process is often called the collapse of the wave function. Actually, only the information transfer is suddenly interrupted. The wave packet does not collapse but it slowly disappears in the ocean of fluctuating waves and the particle immediately starts to build up a new wave accompanying it.

Extended molecules such as $C_{60}$ buckyballs [19] also show the well-known interference pattern after being diffracted, for instance, by a double-slit aperture. Obviously, the phase coupling condition can also be applied to molecules. If a molecule and its accompanying wave are at rest, the wave shows no modulation in space and all atoms oscillate in phase. From the viewpoint of a moving observer, the wave exhibits a modulation in the direction of the relative motion and the oscillating atoms are no longer in phase because the plane of simultaneity is tilted. Nonetheless, the phase matching holds for each individual atom.

## 5. Formation of Waves Accompanying Massive Particles

After having discussed the close coupling of the particle and accompanying wave, the crucial question is: where does the wave come from and how is it possible that each particle with non-zero rest mass is accompanied by a wave?

The escort wave cannot be supplied out of the energy reservoir of the particle. The energy conservation demand can only be satisfied when the particle synchronizes the vacuum fluctuations nearby by adjusting the phases of the fluctuations. The phase adjustment of the vacuum fluctuations by particles with non-vanishing rest mass is the base of all quantum mechanical phenomena.

All spontaneous ordering processes are energy-driven. Therefore, it stands to reason that the formation of ordered waves accompanying massive particles is also energy-driven.

With the help of statistical mechanics, the physical phenomena of multi-particle systems can be described. A multi-particle system strives for a state with a minimal free energy *F*. Thereby, the entropy *S* which is a measure for the disorder of a system must not decrease. A reduction in the energy is usually combined with the reduction in the disorder. However, when the wave–particle system includes the abundance of non-synchronized vacuum fluctuations, the entropy may nonetheless slightly increase.

The presence of a massive vibrating particle is a premise for the formation of an ordered wave. The frequency of the wave Ω is fixed to the mass of the particle *m* by the relation $\hbar \Omega =mc^{2}$. The assertion that the wave–particle system lowers its free energy when the phases of the fluctuating waves are adjusted to each other cannot be proven here. Hopefully, specialists on statistical physics can confirm this hypothesis in the near future. It is certain that the calculations must also consider special relativity.

A particle at rest and the wave accompanying it would be in equilibrium if the amplitude of the wave and its phase did not depend on space coordinates. This status can never be fully achieved. The equilibrium is always disturbed by the interaction with vacuum fluctuations. It may be helpful to define a near, an intermediate and a remote zone. In the near zone, the amplitude and phase of the oscillations practically do not depend on space coordinates. In this region, the wave is nearly coherent. In the far zone, there are only stochastically fluctuating waves. In the intermediate zone, vacuum fluctuations are absorbed in the structured escort wave if the frequency and wave vector nearly agree with the frequency and wave vector of the already stabilized escort wave. The integration of new vacuum fluctuations continually causes distortions of the already fairly well-organized escort wave. As the total energy and total momentum of the escort wave are much larger than the energy and momentum of newly incorporated vacuum fluctuations, the fusion will only cause slight modifications of the escort wave and of the particle’s trajectory.

## 6. Why Does the Schrödinger Equation Yield Proper Results?

Many problems in atomic physics concern the motion of particles in time-independent electric potentials. As a consequence, the total energy of the particle has the form
(10)$$E=mc^{2}+\frac{p^{2}}{2m}+V\left(r\right)=mc^{2}+\frac{p^{2}}{2m}+V\left(r\right)=mc^{2}+E_{kin}+E_{pot}.$$
As the kinetic energy is expressed by the squared momentum term, the direction of the particle’s velocity is not relevant. Thus, the kinetic energy is equal for the particle moving in either direction.

Schrödinger aimed to formulate the energy conservation condition for a wave governing the kinematics of the particle. For their consideration, the character of the wave is not relevant. Therefore, the Schrödinger equation can not only be applied for the virtual wave function but also for the real escort wave. The energy conservation law of the escort wave emerges from the energy conservation law of particle (10) when energy *E* and momentum p of the particle are replaced by the energy E=ℏω and momentum p=ℏk of the wave. The energy and momentum of the wave are extracted from the escorting wave by applying the operators
(11)iℏ∂/∂tand−iℏ∇.
However, remains to be explained why the energy conservation equations for the particle and accompanying wave contain the same potential energy term V(r).

This is not a trivial question. In the case of the hydrogen atom, the wave function must be neutral. Otherwise, one would observe shielding effects. This means that the Coulomb potential can have no direct influence on the wave function. In the case of the harmonic oscillator, the problem is much more sophisticated. The total energy of diatomic molecules depends on the configurations of all molecular electrons (at least those of the valence shells). If one wants to determine the potential curves, the configurations of the electrons must be calculated by sophisticated (quantum mechanical) calculations for all internuclear distances. The space-dependent force equally acting on the two atoms is derived from the total energy of the molecule as a function of the internuclear distance.

Neither in the case of the hydrogen atom nor in the case of the harmonic oscillator can the wave directly obtain the information on the potential. Actually, the information on the particle’s kinematics can only be transferred to the wave via the phase-coupling mechanism. Caused by the tilting of the simultaneity plane, the local momentum of the particle is encoded in the de Broglie wave length of the wave. This mechanism only works when the particle really follows (stochastically modified) classical trajectories. Only Louis de Broglie [11] has proven that the wave obtains the information on the potential via the phase coupling mechanism. This consideration has not been taken up again.

After having replaced the energy and momentum of a particle by the corresponding operators and after having realized that the particle and accompanying wave are subjected to the same potential, the energy conservation law expressed for particles by Equation (Equation 10) corresponds to the energy conservation law for the escorting wave Ψ(r,t)
(12)$$i\hbar \frac{\partial }{\partial t}\Psi \left(r,t\right)=\left(mc^{2}-\frac{\hbar^{2}}{2m}\Delta +V\left(r\right)\right)\Psi \left(r,t\right),$$
Except for the rest energy term $mc^{2}$, this equation is equal to the time-dependent Schrödinger equation. When the potential does not depend on time, the wave Ψ(r,t) can be written in the form $\psi (r)e^{-i\omega t}$. The probability flux of a traveling wave is given by the wave intensity ${\left|\psi \left(r\right)\right|}^{2}$ times the group velocity $v_{g}$. As the probability flux must be continuous, the wave amplitude ψ(r) is space dependent.

After replacing the term iℏ∂Ψ(r,t)/∂t in Equation (Equation 12) by $\hbar \omega \psi \left(r\right)e^{-i\omega t}=E\psi \left(r\right)e^{-i\omega t}$, the time dependence factor $e^{-i\omega t}$ can be omitted on both sides of the equation. Finally, the time-independent eigenwert equation has the form
(13)$$\left(E-mc^{2}\right)\psi \left(r\right)=\left(-\frac{\hbar^{2}}{2m}\Delta +V\left(r\right)\right)\psi \left(r\right)=H\psi \left(r\right)$$
with the Hamilton-operator *H*. This equation is identical to the time-independent Schrödinger equation. As has been argued for the particle, the squared momentum term representing the kinetic energy of the wave is equal for the wave moving in either direction. This means that the kinetic energy of the wave is also given correctly for standing waves.

One can only understand the basic mechanism of quantum mechanics when a frequency is attributed to the wave function. As the wave function does not depend on time, one cannot comprehend the phase coupling of the particle and wave. In general, the frequency of the matter wave ω=E/ℏ is several orders of magnitude larger than the frequency usually associated with the Schrödinger wave function $\omega_{{}_{S}}=\left(E-mc^{2}\right)/\hbar$. The omission of the constant rest energy term does not affect the energy differences and the transition probabilities between different quantum states. Therefore, the Schrödinger equation provides correct results for practically all quantum phenomena.

However, the omission of the rest energy term has serious logical consequences. The wave has lost its invariance with respect to Lorentz transformations and the de Broglie wave length can no longer be derived from special relativity. If the de Broglie wave length λ=h/(γmv) is simply introduced as an empirical relation coupling the wave length of the wave function to the momentum of the particle, the assertion that quantum mechanics is based on special relativity is concealed. The compliance of the energy and the momentum of the particle and wave is lost. The particle and wave no longer move in unison. This may be the reason why, until now, the wave function could not be associated with a real wave and has been introduced as a purely mathematical tool.

In the introductory sections of quantum mechanics text books, one should use the time-dependent Equation (Equation 12) instead of the time-independent Schrödinger Equation (Equation 13). On the basis of this equation, one should explain the relativistic interplay of massive particles and the real matter wave packets accompanying them. First of all, it is important to explain the formation of the escorting wave and the phase coupling of particle and matter wave in order to understand how the wave obtains the information on the kinematics of the particle. Moreover, it should be made clear that the de Broglie wave length emerges from the relativistic tilting of the simultaneity plane. Only after the escort wave concept of quantum mechanics has been fully understood can one go over to the time-independent Schrödinger equation in the subsequent sections of the text books.

## 7. Diffraction and Scattering Processes

When a stream of identical particles hits an aperture, the probability of finding particles far behind the aperture shows a characteristic diffraction pattern. The well known diffraction pattern of a double slit aperture is shown in Figure 2. The wave length associated with the particle is the de Broglie wave length. The same intensity distribution emerges when the experiment is performed with the light of the same wave length. Obviously, massive particles obey wave laws. The interference pattern of a single slit is equal to the envelope function. The double slit interference causes the fragmentation of the broad peaks of the envelope function into several narrow peaks.

The probability density depicted in Figure 2 comes from the interaction of particle and escorting wave. The trajectory of the particle and the structure of the wave influence each other. Sometimes, the information flux may be predominantly directed from the particle to the wave. Sometimes, it also may be predominantly directed from the wave to the particle. Depending on the circumstances, one of the two mechanisms prevails. Before reaching the aperture, the particle is moving on a straight line. Thus, the escort wave takes on the form of a plane wave. At the double slit aperture, the plane wave is diffracted according to Huygens’s principle. Behind the aperture, the diffracted wave guides the particle.

Just behind the aperture, the wave still has a quite uniform intensity distribution. In this region, the stochastic influence of the vacuum fluctuations leads to a broad probability density. The spreading of the probability density is comparable with the diffraction pattern of a single slit. With increasing distance from the aperture, the diffracted wave develops a more pronounced intensity distribution. In the Fresnel diffraction zone, the wave intensity at the minimum locations is still larger than zero. In this region, it is still possible that particles change from one peak to an adjacent one. This is no longer possible in the Fraunhofer diffraction zone, where the minima reach zero intensity. Far from the aperture, the particles move straight along the ridges of the intensity distribution. The velocity of the particles agrees with the radial group velocity of the escort wave. This means that the particle and associated wave packet reach the detection plane at the same time.

As the free energy of the wave–particle compound reduces if the particle enters regions with higher wave intensity, the particle will tend to enter these high-intensity regions. The stochastic velocity component prevents all particles piling up in the intensity maxima. From the experimental data, one can conclude that the probability of finding the particle will be proportional to the wave intensity. Thus, the probability wave hypothesis of Born holds for diffraction processes.

Figure 2 does not show the intensity of the matter wave. Just behind the aperture the wave intensity quickly decreases. The particle is stochastically embedded in one of the rays of the diffracted wave. On its way to the detector plane, the particle will complement its escort wave by building up a new plane wave. This plane wave has no guiding function. Only the sparse diffracted wave has a weak but still non-vanishing steering influence.

The same considerations hold for scattering processes. The incoming particle is accompanied by a plane wave. Due to the interaction with the scattering object, an outgoing spherical wave arises. As in the case of diffraction processes, the probability density of finding the scattered particle is proportional to the intensity of the scattered spherical wave, thus again confirming Born’s probability wave hypothesis.

## 8. Harmonic Oscillator

In contrast to the unidirectional motion of free particles, bound particles move back and forth. As a consequence, the accompanying wave has to have two components with opposite group velocities. The particle is in phase with the partial wave traveling in the same direction. The traveling wave with opposite group velocity does not interact with the particle because the Doppler frequency shift prevents the phase coupling of the particle and wave.

In general, two atoms attract each other at large distances. At low distances, they repel each other. In the neighborhood of the equilibrium distance, the potential can be approximated by a harmonic oscillator potential $V\left(x\right)=kx^{2}/2$ with the spring constant *k*. Under the assumption that one of the atoms is much heavier than the other one, the heavy atom is practically fixed in space. By deriving the total energy of the molecule with respect to the internuclear distance, one obtains the space-dependent force which accelerates or decelerates the light atom. Thus, the amplitude and the frequency of the atomic vibration are well-defined. In contrast, the escorting wave can only obtain the information on the potential from the kinematics of the light atom.

In the Minkowski diagram of Figure 3, the trajectories of a particle and wave fronts of the escorting wave are shown for the n=3 state of the harmonic oscillator. The steeply rising curves represent four possible trajectories of the particle. Together, they form a full harmonic vibration.

The gently rising and gently falling continuous curves indicate the wave fronts of the traveling waves associated with the back and forth motion of the particle. Adjacent wave fronts differ by $c\tau^{*}$. The dashed curves in between are wave fronts with opposite phases. The gently rising wave fronts are associated with the partial wave moving to the right. Accordingly, the gently falling wave fronts belong to the partial wave moving to the left. In fact, an infinite number of possible trajectories can be drawn in the figure because each site of the base line can be the starting point of two trajectories, with one to the left and one to the right. All these trajectories would be in phase harmony with the wave.

As can be seen in the Minkowski diagram of Figure 1, the slope of the trajectory of a particle moving with the velocity *v* in the laboratory system is c/v and the slope of the associated wave front is v/c. This means that the trajectory and associated wave fronts have reciprocal slopes. This relationship persists when the particle is accelerated. Thus, at each crossing of a trajectory with an associated wave front, the two curves have reciprocal slopes. In Figure 3, these crossings are indicated by small circles.

The interference of the two traveling waves leads to a standing wave. The three continuous vertical lines in Figure 3 indicate the positions of the nodes and the four dashed vertical lines indicate the positions of the antinodes.

It makes sense to compare the intensity of the matter wave with the probability of finding the particle at a given location. The movement of a harmonically vibrating particle is described by
(14)$$x\left(t\right)=x_{{}_{0}}\sin\left(\omega_{{}_{V}}t\right)$$
with the angular frequency of the molecular vibration $\omega_{{}_{V}}=2\pi /T_{{}_{V}}$. For a given amplitude $x_{{}_{0}}$, the energy of the particle is $E=kx_{{}_{0}}^{2}/2$.

When Δt indicates how long the particle stays in the interval [x,x+Δx], the classical probability C(x) of finding the particle in this interval is given by
(15)$$C\left(x\right)\Delta x=\frac{\Delta t}{T_{{}_{V}}/2}.$$
Using the differentials dt and dx instead of Δt and Δx, Equation (Equation 15) can be written in the form
(16)$$dx/dt=\frac{2}{C\left(x\right)T_{{}_{V}}}.$$
Differentiated with respect to *t*, Equation (Equation 14) has the form
(17)$$dx/dt=x_{{}_{0}}\;\omega_{{}_{V}}\cos\left(\omega_{{}_{V}}t\right)=\;2\;\pi \sqrt{x_{{}_{0}}^{2}-x_{{}_{0}}^{2}\sin^{2}\left(\omega_{{}_{V}}t\right)}/T_{{}_{V}}=2\pi \sqrt{x_{{}_{0}}^{2}-x^{2}}/T_{{}_{V}}.$$
The comparison of Equations (16) and (17) leads to the classical probability density
(18)$$C\left(x\right)=\frac{1}{\pi \sqrt{x_{{}_{0}}^{2}-x^{2}}}$$
with weak singularities at the turning points.

The probability of finding the particle between the two turning points $-x_{{}_{0}}$ and $+x_{{}_{0}}$ is equal to unity, as can be concluded from the integral
(19)$$\int_{-x_{{}_{0}}}^{+x_{{}_{0}}}C\left(x\right)dx=\frac{1}{\pi }\int_{-x_{{}_{0}}}^{+x_{{}_{0}}}\frac{1}{\sqrt{x_{{}_{0}}^{2}-x^{2}}}dx=\frac{1}{\pi }\arcsin\left(x/x_{{}_{0}}\right){|}_{-x_{{}_{0}}}^{+x_{{}_{0}}}=1.$$

Experimentally, it is not possible to determine the probability density of a bound particle. A scattering experiment only provides the absolute square of a matrix element, which contains the wave functions of the initial and final states in addition to the transition operator. Thus, one does not test the probability density but the wave function of the examined molecule.

If the trajectories are deterministic, the probability density outside the turning points is equal to zero. This behavior is not adequate, because the tunnel effect proves that the probability density of particles beyond the turning points is not equal to zero. The tunneling probability decreases exponentially with the width and the height of the tunnel barrier. Therefore, in the case of the harmonic oscillator, the probability density must also exponentially decrease outside the classical turning points. This finite probability density can only be caused by the disturbance of the particle’s trajectory by the vacuum fluctuations. Here, this stochastic effect is approximately accounted for by convoluting the classical probability density C(x) with the normal distribution N(x), which is characterized by the standard deviation σ
(20)$$N\left(x\right)=e^{-{\left(x/\sigma \right)}^{2}/2}/\left(\sqrt{2\pi }\sigma \right).$$
The more realistic probability density W(x) resulting from the convolution of the classical probability density and the normal distribution is given by the integral
(21)$$W\left(x\right)=\left(C*N\right)\left(x\right)=\int_{-\infty }^{+\infty }C\left(x-x^{\prime }\right)N\left(x^{\prime }\right)dx^{\prime }.$$
The standard deviation of the normal distribution σ was chosen to be $0.65\;c\tau^{*}$ in order to correctly describe the tails of the probability density distribution. Escort wave and wave function optimally fit together when the energy *E* of the particle is chosen to be $3\;\hbar \omega_{{}_{V}}$ with the vibrational frequency $\omega_{{}_{V}}$. On the other hand, the energy of the n=3 quantum state is $3.5\;\hbar \omega_{{}_{V}}$. It is quite likely that the energy mismatch of $0.5\;\hbar \omega_{{}_{V}}$ has its origin in the stochastic velocity component of the particle.

The resulting probability density of finding the particle is shown in Figure 4 by a dashed curve. The intensity of the n=3 wave function is shown by a continuous curve. The two curves are clearly different. The wave function squared has three zeros whereas the probability density of finding the particle has no zeros at all. The periodicity of the wave function squared with the de Broglie wave length emerges from the interference of the counter-propagating traveling waves. The interference effect does not affect the probability density of the particle. The striking difference of the wave function squared and the probability density proves that the Copenhagen interpretation of the wave function is not adequate for excited quantum states.

In Figure 5, Schrödinger’s wave function (continuous curve) and the approximated space dependence of the escort wave (dashed curve) are compared. The modulus of the escort wave was chosen to be equal to the square root of the probability density shown by the dashed curve in Figure 4, whereas the phase shift is taken from the space-dependent de Broglie wave length of the particle. The excellent agreement of the two curves proves that convoluting the classical probability density with a normal distribution provides a good approximation of the probability density. It has to be emphasized that the space dependence of the escort wave shown by the dashed curve in Figure 5 was exclusively derived by means of classical and statistical physics. Probably, the two curves in Figure 5 will fully agree when the exact probability density is determined by averaging over all stochastically modified trajectories. The agreement of the wave function with the space dependence of the escort wave, which is solely deduced from classical considerations, supports the assumption that massive particles are always strictly localized and move on continuous trajectories.

The stochastic variation of the particle’s velocity is highly relevant for the shape of the ground state wave function. Without the influence of vacuum fluctuations, a particle in ground state would permanently stay at rest in the potential minimum and the energy of the ground state would be zero. Actually, this equilibrium is disturbed by the interaction with vacuum fluctuations. Edward Nelson [20] derived the formalism of quantum mechanics from the statistical mechanics of point particles without presuming the existence of waves. After having determined the density distribution of particles, Nelson formally derived a wave which was equal to the ground state wave function. Thus, for ground states, statistical mechanics provides the probability densities and the ground state energies known from quantum mechanics. The considerations of Nelson are restricted to the ground state because the wave-function squared is non-zero everywhere. For excited states, this precondition is not fulfilled when the probability density is assumed to be proportional to ${\left|\psi \right|}^{2}$. In the escort wave concept, the probability density is smooth and has no zeros. Nelson’s consideration can also probably be applied to excited states when the probability density of the escort wave concept is used.

## 9. Atomic Hydrogen

The quantum states of atomic hydrogen are characterized by three quantum numbers, namely the principal, the orbital and the magnetic quantum numbers n,l and *m* with n>l≥|m|, respectively. The wave functions associated with the quantum states of atomic hydrogen have been extensively discussed in text books. However, the escort wave concept requires that the electron orbits are also considered. As the Coulomb potential is mathematically equivalent to the gravitational potential, the classical orbits must comply with Kepler’s laws of planetary motion. First, the orbits are ellipses with one of the focal points in the center of the 1/r- potential. Second, for each orbit the angular momentum is constant. Third, the squares of the circulation periods behave like the cubes of the semi-major axes of the orbits. Kepler’s laws are exactly fulfilled for atomic hydrogen. For hydrogen-like atoms with single electrons in the valence shell, they are still approximately valid. However, if several electrons are circling the nucleus in the valence shell, the electrons strongly interact with each other. Thus, the correlated trajectories distinctly deviate from the elliptical shape.

Because of its charge, the electron is subjected to the Coulomb potential. An extended wave carrying a distributed charge would shield and thus modify the Coulomb potential. Such a shielding effect has never been observed. Therefore, the escort wave must be neutral. Indeed, a neutral wave cannot be influenced by the Coulomb potential. This conflict can only be solved by assuming that the wave obtains information on the potential from the kinematics of the particle via the phase-coupling mechanism.

For l=0, the electron orbits degenerate into straight lines. Planets or comets moving in the solar system with exactly zero orbital momentum would hit the sun on the first pass. In the case of atomic hydrogen, such trajectories exist because the electron will not react with the proton. Moreover, the electron will normally not hit the tiny proton because its trajectory is always disturbed by the interaction with vacuum fluctuations.

In contrast with classical physics, the orientation of consecutive electron orbits will incessantly change due to the stochastic velocity component. For l=0, this misalignment of consecutive orbits may be quite distinct, because electrons are strongly accelerated in the vicinity of the nucleus. This acceleration sensitively depends on small deviations of the trajectory. On average, the electron will cover all possible directions leading to a spherically symmetric probability density.

For l=0, the escort wave is a standing wave, which consists of two spherical waves traveling inwards and outwards. The electron is only in phase with one of the traveling waves. If the electron’s velocity is opposite to the group velocity of the traveling wave, the Doppler shift prevents the electron and wave from interacting. As the amplitudes of the counter-propagating waves are counter-rotating in the complex plane, the sum of the two traveling waves has the same phase everywhere. This phase can be arbitrarily chosen to be zero. Analogously to the harmonic oscillator case, the intensity varies between ${\left|\psi \left(r\right)\right|}^{2}$ and zero, where |ψ(r)|/2 is the common amplitude of the traveling spherical waves. The standing wave has n−1 nodes. The probability of finding the electron is reciprocally proportional to the electron’s velocity. As the trajectories of the electron are continuous, there cannot be spheres with vanishing probability density. As a consequence, Born’s probability hypothesis must be wrong for excited *s*-states.

For l>0, the electron moves on elliptical orbits. The attractive Coulomb potential is modified by a repulsive centrifugal term $L^{2}/\left(2mr^{2}\right)$ with the angular momentum squared $L^{2}=l\left(l+1\right)\;\hbar^{2}$. With the increasing orbital quantum number *l*, the eccentricity of the elliptic orbits decreases. The number of spherical node surfaces in the radial intensity distribution reduces to n−l−1.

Without an external magnetic field, the electron orbits continually change their orientation due to the interaction with vacuum fluctuations. Therefore, on average, the electron density is spherically symmetric over time. By applying an external magnetic field, the *z* axis of the spherical coordinate system is aligned in parallel to the field direction, thus leading to a cylindrically symmetric probability density.

For m=0, the inclination angle of the elliptical orbit is equal to $90^{\circ }$. The electron periodically moves from the north pole region to the south pole region and vice versa. With an increasing |m|, the inclination angle, which is equal to the reversal angles $\vartheta_{max}$ and $-\vartheta_{max}$, decreases. For |m|=l, the electron would move exactly in the equatorial plane if the stochastic velocity component were disregarded. The sign of *m* indicates the rotational direction.

As long as l>|m| there are two partial waves with opposite group velocities in the direction of the polar angle ϑ which together form a standing wave with l−m nodes. Here, the escort wave concept is also in explicit contradiction to the Copenhagen interpretation. The probability density of finding the electron is not zero when the wave intensity is zero.

With respect to the azimuthal angle φ, the motion of the electron is unidirectional. Therefore, the φ-component of the wave function $e^{im\varphi }$ is complexly valued without any zero intensity surfaces. Due to the conservation of the angular momentum $mv{\;}_{⊥}\;r$, the tangential angular velocity $v{\;}_{⊥}$ of the electron changes with its distance from the proton. As the de Broglie wave length is reciprocally proportional to the velocity of the electron, the tangential component of the de Broglie wave length is proportional to the electron–proton distance. This means that, with respect to the wave function, it does not matter where the aphels and perihels of the orbits are located or, in other words, the orientation of the semi-major axis of the elliptic orbits has no influence on the angular dependence of the escort wave.

The electron on its elliptical orbit induces a magnetic moment which causes a slow precession of the elliptical orbits with respect to the magnetic field. This precession leads to a fine structure splitting of the magnetic sublevels, but does not significantly change the structure of the escort wave.

In the harmonic oscillator case, it was not possible to determine the actual position of the light atom on its way back and forth. This information gap enlarges for movements in three dimensions. Not only is the location of the hydrogen electron unknown, but so is its actual trajectory. For a given quantum state, the electron can move on an infinite number of trajectories.

A continuous trajectory is a necessary condition for the formation of the matter wave. Otherwise, the wave does not obtain the information about the potential. Obviously, both the trajectory and matter wave are real and indispensable elements of microphysics in order to understand the quantum mechanics. The escort wave concept does not substantially enlarge our knowledge about the electron’s movement. However, it is now irrefutable that microscopic particles move on continuous approximately classical trajectories.

## 10. Comparison of Bohmian Mechanics and Escort Wave Concept

The escort wave guides the associated particle. In Bohmian mechanics [17,18,21], the pilot wave, originally proposed by Louis de Broglie [12], also has a guidance function. Therefore, it is necessary to discuss the differences between the two concepts. Bohm transformed the Schrödinger equation into a hydrodynamic equation. He concluded that the flux distribution of the wave function locally agrees with the flux distribution of the associated Bohmian trajectories.

The flux is non-zero when the traveling wave associated with the moving particle is complexly valued as in the case of the φ-dependence of the atomic hydrogen wave functions with m>0. In this case, Bohmian mechanics rightly postulates that the electron is moving on a circular orbit with respect to the azimuthal angle φ.

The main problem of Bohmian mechanics is the fact that the probability flux of real-valued wave functions is zero everywhere. This is the case for the harmonic oscillator and for atomic hydrogen wave functions with zero magnetic quantum numbers. Therefore, particles described by a real-valued wave function are assumed to be at rest. As a consequence, the electron must be prevented from falling down to the nucleus by introducing a non-local quantum potential. The unrealistic properties of Bohmian trajectories have already been discussed in the literature [22,23].

In sharp contrast to Bohmian mechanics, the escort wave concept interprets the real-valued standing wave as a superposition of two waves traveling in opposite directions. Thus, the flux components of the two traveling waves add up to zero. Summarizing the movements in the radial, polar and azimuthal directions, the hydrogen electron moves on elliptic orbits. Thus, the electron orbits in the Coulomb potential are stabilized by the centrifugal force. It is not necessary to postulate quantum potential. The electron orbiting the proton and the planets circling around the sun move on identical orbits. However, the electron orbits are quantized with respect to the energy and angular momentum because the escorting wave has to fulfill boundary conditions. Thus, the presence of the escorting wave does not distinctly modify the classical orbits, but enforces quantization effects typical for microphysics.

The most striking advantages of the escort wave concept are that massive particles approximately move on classical trajectories and that the wave function is equal to the space dependence of a rea wave accompanying the particle.

There is another argument as to why Bohmian mechanics is in conflict with experimental findings. If atomic electrons would be at rest, neutral atoms would be associated with strong dipoles. The interaction of two dipoles is much stronger than usual internuclear forces. For atom–atom collisions, such dipole–dipole interactions were not observed. In contrast to Bohmian mechanics, the escort wave concept always predicts orbits circling the nucleus with high frequency. Thus, on average, atoms have no net dipole moments over time.

## 11. Character and Structure of Escort Waves

As the wave function is commonly assumed to be a purely mathematical tool without a concrete physical counterpart, the character and the structure of the escort wave were not examined until now. Hence, the considerations in this section are rather speculative.

The transition from a quantum state to a lower one always leads to the emission of a photon. On the other side, the absorption of a photon leads to the excitation of a quantum state. In both cases, the photon energy is identical to the energy difference of the two quantum states involved. The strong linkage of photons and quantum transitions suggests that the escort wave is also an electromagnetic wave. However, photons and matter waves behave differently. Whereas photons in free space have no rest system and are always traveling with the speed of light, matter waves have a rest system and their group velocities are always smaller than the speed of light.

Electromagnetic waves with a rest system were only observed when there are metallic or dielectric surfaces nearby. One can hope that vibrating particles also support the formation of stationary electromagnetic waves. Until now, such systems have not been studied. The escort wave concept of quantum mechanics assumes that particles with mass *m* are executing periodic oscillations with the frequency $\Omega =mc^{2}/\hbar$. From the Maxwell equations, one can conclude that the matter wave must also exhibit a modulation in space with the Compton wave length $\lambda_{{}_{C}}=h/\left(mc\right)$. Except for the Compton effect itself, this spatial modulation seems to induce no significant effect, probably because the vibrational amplitude of the particle is comparable with the Compton wavelength. In contrast, there are countless observable effects associated with the de Broglie wave length λ=h/(γmv)≈h/(mv).

Excited atoms principally decay to lower quantum states by emitting photons. Therefore, these atomic states are not in equilibrium with the vacuum state. Actually, the emission rate along a decay chain from a Rydberg state to a low lying excited state is comparable with the emission rate derived from Larmor’s formula [24]. Sometimes, it is argued that electrons cannot move on Kepler orbits because accelerated electrons irradiate energy and thus destabilize the electron orbits. De facto, the emitted radiation is as large as predicted by Larmor’s formula. However, the energy is emitted in quantum form. Obviously, the presence of the escorting wave stabilizes the electron orbit for a while, until after all the quantum state decays. Only the ground state with its stochastic movement is in equilibrium with the vacuum fluctuations.

The field configurations of matter waves are probably similar to the configurations of the electromagnetic fields in resonant cavities and in wave guides at the cutoff frequency. Light waves also comprise waves with circular energy flux. Thereby, one has to discriminate orbital and spin angular momenta [25]. In the case of orbital angular momenta, the orientation of the fields is rotating. In the case of spin angular momenta, there are helical wave fronts. Such helical waves can be generated by spiral phase plates. As these waves carry spin and angular momenta, the escort waves of particles with non-zero spin are most probably of this type. The relative phases of oscillating electric and magnetic fields must be different for different spins. At least the Poynting vector must continually change its direction. However, on average, the energy flux associated with a particle at rest must be zero over time.

The spin is a property not only of the particle but also of the escorting wave. Otherwise, one cannot understand Pauli’s exclusion principle, which says that two electrons with the same spin cannot occupy the same quantum state. Obviously, the presence of a second electron with an identical set of quantum numbers destroys the symmetry of the wave escorting the first electron and vice versa. In contrast, many bosons can be in the same quantum state. At very low temperatures, the escorting waves of an ensemble of identical bosons coalesce into a more extended matter wave, thus forming a Bose–Einstein condensate with a common wave function.

One could speculate that particles with spins combine a vibration in a freely chosen direction with a circulation in the plane perpendicular to it. Such behavior would explain why the magnitude of a spin is distinctly larger than its maximal *z* component. This means that the spin is somewhat wobbling around the external magnetic field. A wave function associated with a half-integer spin particle changes its sign when the particle is rotated by $360^{\circ }$. It is not clear what this assertion means with respect to the escort wave. One could speculate that the vibrational frequency is twice as large as the rotational frequency. Thus, both the vibrational and the rotational components are only equal when the particle is rotated by $720^{\circ }$.

The escort wave is built up very slowly. At the beginning, the matter wave still exhibits the irregularities of the stochastically modified trajectories. Only after the particle has run through many orbits, the escort wave is stable. It no longer shows stochastic anomalies.

## 12. Concluding Remarks

It has to be emphasized that the considerations in this article only concern the interpretation of the wave function and have absolutely no consequences on the mathematical formalism of quantum mechanics.

The wave–particle duality is no longer mysterious. The fact that massive particles are escorted by extended waves explains why their propagation complies with wave laws. In fact, the escort wave concept embeds quantum mechanics into the traditional disciplines of physics, namely mechanics, statistical mechanics, electrodynamics and special relativity.

The question about the completeness of quantum mechanics posed by Einstein, Podolsky and Rosen in 1935 [26] has not been answered to date. Compared with the escort wave concept, the Copenhagen interpretation of quantum mechanics is not complete because it denies the existence of continuous trajectories. Although Heisenberg’s uncertainty relation prevents one specifying the concrete path of a particle, it is epistemically highly significant to know that particles run through continuous trajectories and that the wave function is equal to the space dependence of a real wave.

## Figures and Tables

**Figure 1 entropy-24-01535-f001:**
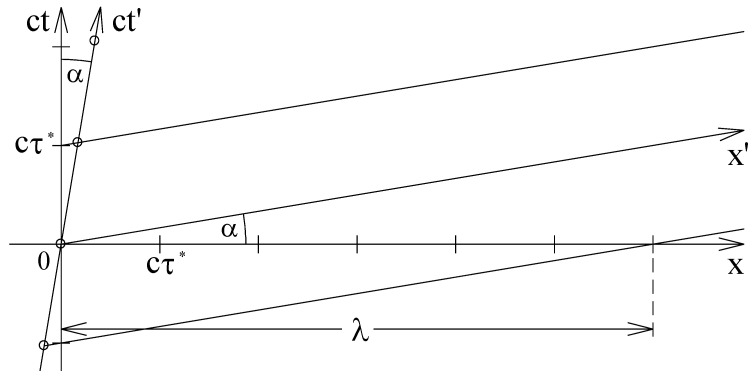
Minkowski diagram of a matter wave being at rest in the reference frame $P^{\prime }=\left(x^{\prime },ct^{\prime }\right)$. The $x^{\prime }$ axis and the two lines parallel to the $x^{\prime }$ axis represent three wave fronts. The phases of two adjacent wave fronts differ by a wave period $c\tau^{*}$. This means that, at the three wave fronts, the wave oscillates in phase. For an observer in the laboratory system P=(x,ct), the matter wave exhibits a spatial modulation. This modulation emerges from the relativistic tilting of the simultaneity plane. The wave length λ of the matter wave is traditionally called a de Broglie wave length.

**Figure 2 entropy-24-01535-f002:**
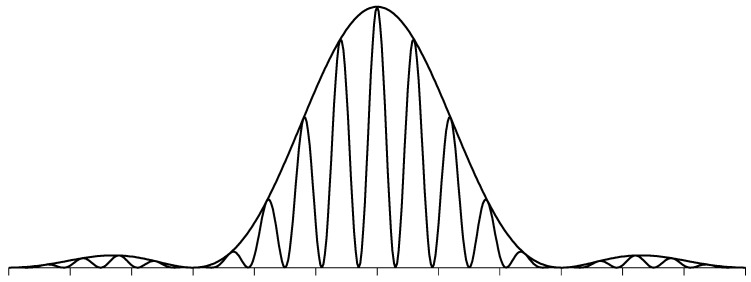
The angular dependence of the probability of finding particles after being diffracted on a double slit aperture. The distance between the center lines of the two slits was chosen to be five times larger than the slit width. The envelope function is equal to the diffraction pattern of a single slit. The horizontal axis is subdivided in angular degree units.

**Figure 3 entropy-24-01535-f003:**
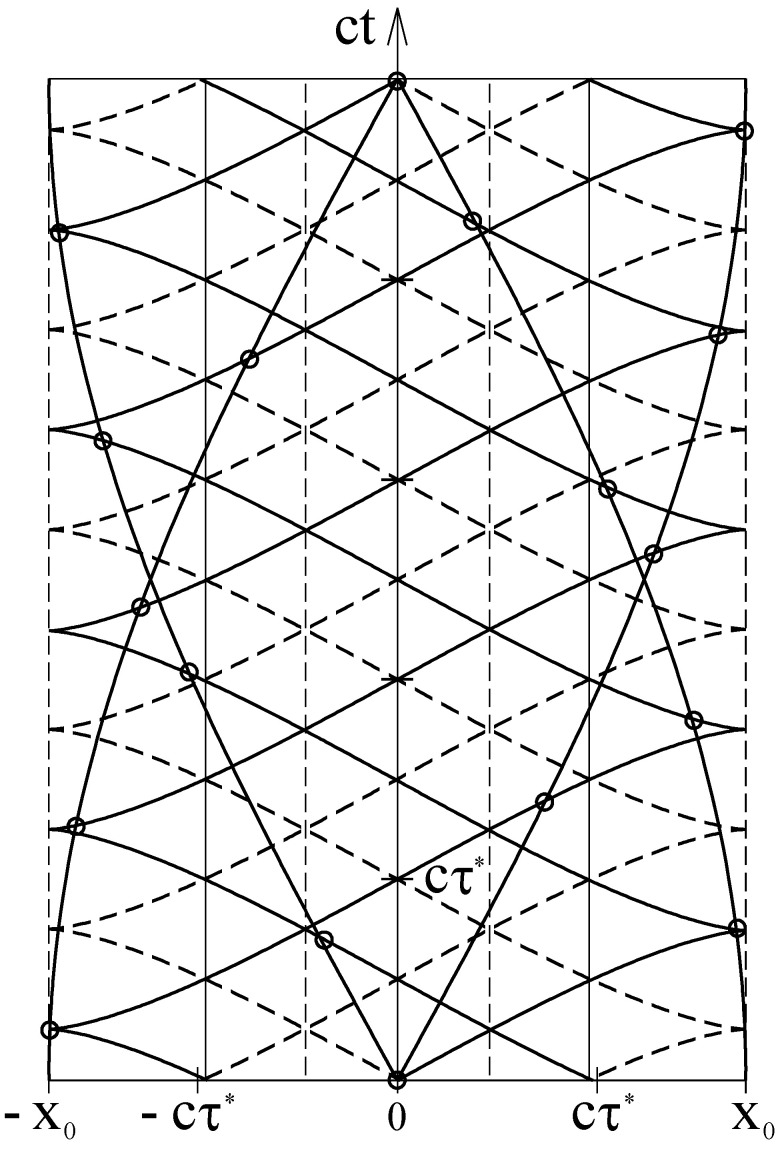
Trajectories of a harmonically vibrating particle and wave fronts of the wave escorting it. Both axes of the Minkowski diagram are subdivided into units of $c\tau^{*}$.

**Figure 4 entropy-24-01535-f004:**
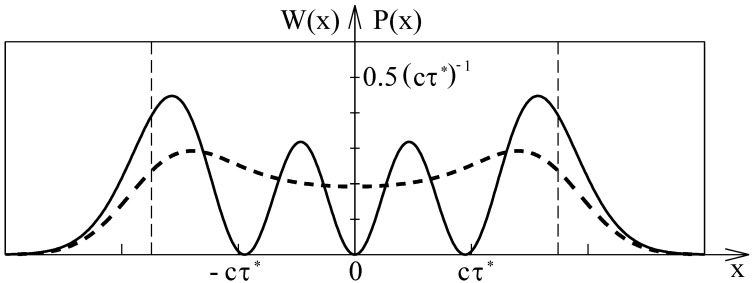
The absolute square of the n=3 state wave function $P\left(x\right)=\psi \left(x\right)\psi^{*}\left(x\right)$ of the harmonic oscillator (continuous curve) in comparison with the corresponding probability W(x) of finding the particle (dashed curve). The stochastic velocity component is simulated by convoluting the classical probability density with a normal distribution. The width of the normal distribution was chosen to be $\sigma =0.65\;c\tau^{*}$. The dashed vertical lines indicate the classical turning points.

**Figure 5 entropy-24-01535-f005:**
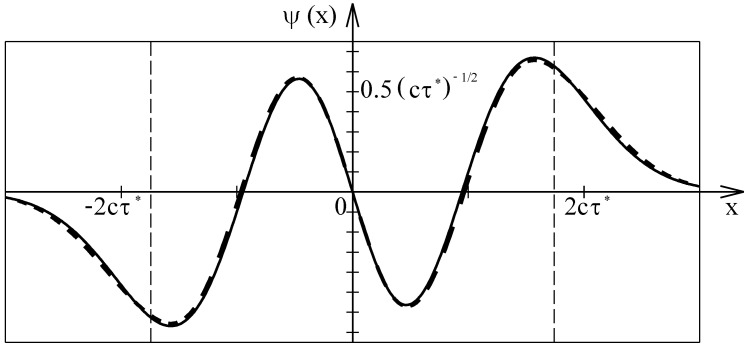
Exact and approximated space dependence of the n=3 state of the wave accompanying a particle moving in a harmonic oscillator potential. The solid curve shows the solution of the Schrödinger equation. The dashed curve shows the approximated space dependence of the escort wave. The modulus of the escort wave is equal to the square root of the approximated probability density shown in Figure 4, whereas the phase is derived from the space-dependent de Broglie wave length. The dashed vertical lines indicate the classical turning points.

## Data Availability

Not applicable.

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
