# Peer review of "A Proposed Interpretation of the Wave–Particle Duality"

_entropy, 2022, doi:10.3390/e24111535_

Round 1
Reviewer 1 Report
This is a strange work. Much of what is presented in it could be used in a quantum mechanics course. However, there is nothing really new. The author, in the conclusions, speaks of "paradox of wave-particle duality", terminology that is not used in physics. With some rewriting the work can be interesting for another journal like the American Journal of Physics. But it certainly is not adequate for Entropy.
Reviewer 2 Report
The author offers a highly original point of view on a subject that is in itself highly controversial and actual. This manuscript will be of real interest to many readers of the journal. It can be published as it is.
I understand the origin of the discrepancy. The author is proposing a new point of view on quantum mechanics, which differs widely from the usual one, although it has its proper logic, as the manuscript itself indicates. Thus it is to be expected that this manuscript find difficulties in its way to the printing. Personally, I consider better to give a chance to new ideas than to suppress them as soon as they appear, negating them their opportunity. The author's proposal is clearly presented already in the abstract:
"...In this article a more convincing description of the wave function is given which embeds quantum mechanics into the traditional fields of physics, thus making new interpretations dispensable. The new concept is based on the idea that each microscopic particle with non-vanishing rest mass is accompanied by a matter wave, which is formed by synchronizing the vacuum fluctuations in the vicinity of the vibrating particle. The vibrations of particle and wave are phase coupled. Particles move on continuous trajectories. By the phase coupling mechanism the particle transfers the information on its kinematics and thus also on the external potential to the wave. The space dependence of the escorting wave turns out to be equal to the wave function."
Reviewer 3 Report
This paper should rejected.
The formulations therein are not new. The formulations of section 5 and 7 are so old, they can be found more or less in Wikipedia (which shows references). You can Google "matter waves" and find that the Wikipedia entry has the RHS of eq. 15 and its connection to the Compton wavelength also made. This includes relativistic formulations and derivations of the phase and group velocity. Equating relativistic energy E=m*c^2 with Planck-Einstein's law E = h*f has been done long ago! What the author's abstract is alluding to, is a pilot-wave theory.
If you inject the quantization rules of eq. (22) into eq. (18), you would get the relativistic Schrödinger equation (which Schrödinger himself obtained long ago before his famous non-relativistic linear formulation which is the author's eq. (23) without the rest energy m*c^2). This is known today as the Klein-Gordon equation which works well for bosons. This paper does not connect De Broglie's matter wave theory with the Schrödinger equation any more fundamentally than what has been done already. The full theory would not appear until Bohm's formulations. Today, the De Broglie-Bohm theory can include the spin-1/2 of electrons, which this author does discuss but does not formulate. This is important because Fermions are the basis of atoms and molecules. However, the De Broglie-Bohm theory was widely deemed unacceptable by mainstream theorists, mostly because of its explicit non-locality. Bell's theorem (1964) was inspired by Bell's discovery of Bohm's work; he wondered whether the theory's obvious non-locality could be eliminated.
Some parts of this paper are totally wrong. E.g. he states "It makes sense to emphasize that the particle must not be point-like". One should never confuse the internal structure of a particle and its wave nature. As far as we can tell from experiments: electrons have no known internal structure i.e. they are point particles. The upper limit for the electron radius is at 10^(-18) meters which is orders of magnitude below its Compton wavelength at 10^(-12) meters or even its "classical electron radius" at 10^(-15) meters. Electrons are treated as point-particles in quantum electrodynamics and the crippling infinities caused by the self-energy of such point particles lead to the Lamb shift which is well verified experimentally. The calculations are addressed via renormalization (tantamount to burying the infinities in the charge or mass of the electron). The author's theory fails completely for the electron and does even begin to address these points in quantum field theory.
The author mentions superconductors. Superconductors can have zero resistance and Superfluids can have zero viscosity thanks to recognized quantum effects at very low temperatures. However, both are often modeled with NON-LINEAR forms of the Schrödinger equation for a collection of particles acting in unison which don't completely obey all the requirements of single-particle quantum mechanics. The author cannot begin to model such systems.
Concerning section 9 on the harmonic oscillator, amongst other things, the somewhat problematic issue of achieving the classical correspondence between the quantum oscillator and the classical oscillator has been worked out. However, it requires a careful treatment in the context of quantum optics. One has to look at the very highly excited states of the quantum oscillator, which are very "ringy" i.e. oscillates at high-frequency as shown somewhat by Figure 4 of the author's paper. Because of the smaller and smaller gaps in energy between these highly excited states, this allows one to look at states within an energy region and to make linear combinations of such states, averaging them out, and removing the "ringy" nature which is needed to achieve the classical limit. To get the peaks of probability of the classical limit, one has to look at asymptotics of the Hermite polynomials, and also through the WKB approximation. This results in integrals and probability distributions not unlike what the author uses. The conventional treatment, though somewhat complicated mathematically, is more comprehensive than that of the author. No pilot theory is even needed for this.
Concerning section 10, the author's quantum description of the hydrogen atom seem to allude mostly to the semi-classical Bohr-Sommerfeld model in which he mentions Kepler states. Note that elliptical Kepler states were ruled out in this semi-classical Bohr theory from the onset. The orbits in Kepler's model case cross each other. This is not the case of electrons in the Bohr model. However, the interpretation of standard quantum mechanics with the Schrödinger equation does not associate any classical pattern to electronic orbits but rather clouds of probability.
In passing we note that Deepak Dhar claimed in his 2016 paper to uncover Kepler states in highly excited Rydberg states. However, he makes alternate use of the Bohr model and standard quantum mechanics. Again, no pilot theory. If Dhar's work is correct, this would be another example of the correspondence to the classical limit.
The author's theory cannot be fundamental. Mass is an important parameter for the author but mass is a parameter which is ON INPUT in relativity and quantum mechanics. In General Relativity (GRT), mass is more fundamental. It appears as a geometric distortion in the fabric of space-time. Had the author derived a pilot-wave theory from GRT, that would have been more helpful.
BTW, the wave-particle duality doesn't bother us so much. Here is a way of dealing with it, attributed to Richard Feynman:
"If you worry about whether or not an electron is a particle or a wave, here is a simplified form of his answer:
1) The electron is a particle, period.
2) The photon is a particle, period.
3) Anything involving a wave-nature comes the wave of probability as expressed by e.g. the Schrödinger equation."
This view is perhaps cynical but it works. :-)
See one of his lectures,
https://www.youtube.com/watch?v=_7OEzyEfzgg
Round 2
Reviewer 1 Report
I have carefully read your reply. I consider it relevant. I think the change of title that you propose is very important. I would suggest that in the introduction it be made clear that some concepts, usually treated in quantum mechanics textbooks, are discussed in the paper only for the purpose of better understanding the motivations of the article.
Reviewer 3 Report
This paper should be rejected again.
The author tried to address the points raised in the first report but still fails to provide a solution to the wave/particle issue.
This second draft simply tries to obfuscate or gerrymander the issues raised in the first report and this second draft is now full of inconsistencies. The author's choice of semantics i.e. 'escort wave' in effect means a pilot wave theory. He has relinquished the notion that particles cannot be point-like in his first draft to now consider a proton as point particle as mentioned in line 300. Protons have internal structure at the level of 1 Fermi. They are NOT point-particles.
He relinquishes the fully relativistically invariant formulation of the first draft (namely what the first report identified as the Klein-Gordon equation) in favor of eq. 5 which is just the non-relativistic Schrodinger equation with an additional rest mass (which is a constant and can therefore be readily absorbed into the eigenvalue). If the author relinquishes relativistic invariance, then why bother with the Minkowski space as shown in Figure 1? Also, the non-relativistic Schrodinger equation is not enough these days, not nearly enough. We need more to account for e.g. quantum field corrections.
As mentioned in the first report, eq. (6) are the quantization rules by which we convert a classical formulation into a quantum wave equation. Deriving the Schrodinger equation takes a bit more that Einstein's light quanta hypothesis: E=hbar*omega and the de Broglie's hypothesis: p =hbar*k. For instance, it can be derived from a comparison of the classical wave equation to the Hamilton-Jacobi equation via the Eikonal equation.
see https://www.sciencedirect.com/topics/computer-science/eikonal-equation
However, the general key quantization rule first given by Born and Jordan is simply to take the Poisson-bracket between 2 classical operators of a given classical Hamiltonian and multiply it by hbar*I where I=sqrt(-1) and provides the canonical commutation rule of the quantum version of these operators.
[x,p ] =I*hbar*{x,p}
See https://en.wikipedia.org/wiki/Canonical_commutation_relation
Nothing in this paper explains that general key rule. The de Broglie-Bohm theory uses a quantum potential inserted into the Hamilton-Jacobi equation which introduces the hbar into the formulation:
https://en.wikipedia.org/wiki/Quantum_potential
from which one gets the Schrodinger equation. This quantum potential acts as a guide of the particle.
The author insists his 'escort' theory is distinct from the de Broglie wave theory but the author provides no general mathematical formulation for it. The author has no counterpart of a quantum potential. At most, he provides it for the harmonic oscillator in a somewhat ad hoc manner. The author's Gaussian distribution of eq. 15 is not really explained and likely works because the ground state of a quantum harmonic oscillator happens to be a Gaussian function! Moreover, the excited states are Gaussians multiplied by Hermite polynomials. Granted, a lot of modeling in quantum chemistry uses combinations of atom-centered Gaussian functions. However, this is because of the mathematical properties of Gaussians like the Gaussian product rule which ensure that all molecular integrals can be solved analytically in closed form. Asymptotically, a wave function is more hydrogen-like exp(-constant*r) rather than a Gaussian. In practice, Gaussians make poor atomic and molecular wave functions but convergence in the mean of e.g. a Hartree-Fock calculation will yield a good approximation of the energy but the resulting wave function will be poor.
The first report was perhaps too kind in its critique on the section of the hydrogen atom. The whole section of atomic hydrogen is ridiculous. It would be considered perhaps if the Bohr-Sommerfeld model was still used. Kepler's laws do not apply to the quantum system of the hydrogen atom. They only exist approximately for certain types of very highly excited states.
This paper claims to have resolved the EPR paradox. It doesn't! and its mystery has little to do with continuous functions or trajectories. Most of the known solutions to quantum mechanics are continuous functions anyway, with a few exceptions necessitating a discontinuity in the wave function at singularities (point-particles) or anomalies. The EPR paradox or problem is an issue of quantum nonlocality and this paper does not address that.
In the end, the author's 'escort wave' theory does not present anything better than the de Broglie-Bohm theory.
